# Decoupling Hierarchical Recurrent Neural Networks With Locally Computable Losses

## Abstract

Learning long-term dependencies is a key long-standing challenge of recurrent neural networks (RNNs). Hierarchical recurrent neural networks (HRNNs) have been considered a promising approach as long-term dependencies are resolved through shortcuts up and down the hierarchy. Yet, the memory requirements of Truncated Backpropagation Through Time (TBPTT) still prevent training them on very long sequences. In this paper, we empirically show that in (deep) HRNNs, propagating gradients back from higher to lower levels can be replaced by locally computable losses, without harming the learning capability of the network, over a wide range of tasks. This decoupling by local losses reduces the memory requirements of training by a factor exponential in the depth of the hierarchy in comparison to standard TBPTT.

## 1 Introduction

Recurrent neural networks (RNNs) model sequential data by observing one sequence element at a time and updating their internal (hidden) state towards being useful for making future predictions. RNNs are theoretically appealing due to their Turing-completeness Siegelmann and Sontag (1995), and, crucially, have been tremendously successful in complex real-world tasks, including machine translation Cho et al. (2014); Sutskever et al. (2014), language modelling Mikolov et al. (2010), and reinforcement learning Mnih et al. (2016).

Still, training RNNs in practice is one of the main open problems in deep learning, as the following issues prevail.

(1) Learning long-term dependencies is extremely difficult because it requires that the gradients (i.e. the error signal) have to be propagated over many steps, which easily causes them to **vanish or explode** Hochreiter (1991); Bengio et al. (1994); Hochreiter (1998).

(2) Truncated Backpropagation Through Time (TBPTT) Williams and Peng (1990), the standard training algorithm for RNNs, requires memory that grows linearly in the length of the sequences on which the network is trained. This is because all past hidden states must be stored. Therefore, the **memory requirements** of training RNNs with large hidden states on long sequences become prohibitively large.

(3) In TBPTT, parameters cannot be updated until the full forward and backward passes have been completed. This phenomenon is known as the **parameter update lock** Jaderberg et al. (2017). As a consequence, the frequency at which parameters can be updated is inversely proportional to the length of the time-dependencies that can be learned, which makes learning exceedingly slow for long sequences.

The problem of vanishing/exploding gradients has been alleviated by a plethora of approaches ranging from specific RNN architectures Hochreiter and Schmidhuber (1997); Cho et al. (2014) to optimization techniques aiming at easing gradient flow Martens and Sutskever (2011); Pascanu et al. (2013). A candidate for effectively resolving the vanishing/exploding gradient problem is hierarchical RNNs (HRNNs) Schmidhuber (1992); El Hihi and Bengio (1996); Koutnik et al. (2014); Sordoni et al. (2015); Chung et al. (2016). In HRNNs, the network itself is split into a hierarchy of levels, which are updated at decreasing frequencies. As higher levels of the hierarchy are updated less frequently, these architectures have short (potentially logarithmic) gradient paths that greatly reduce the vanishing/exploding gradients issue.

In this paper, we show that in HRNNs, the lower levels of the hierarchy can be decoupled from the higher levels, in the sense that the gradient flow from higher to lower levels can effectively be replaced by locally computable losses. Also, we demonstrate that in consequence, the decoupled HRNNs admit training with memory decreased by a factor exponentially in the depth of the hierarchy compared to HRNNs with standard TBPTT. The local losses stem from decoder networks which are trained to decode past inputs to each level from the hidden state that is sent up the hierarchy, thereby forcing this hidden state to contain all relevant information. We experimentally show that in a diverse set of tasks which rely on long-term dependencies and include deep hierarchies, the performance of the decoupled HRNN with local losses is indistinguishable from the standard HRNN.

In summary, we introduce a RNN architecture with short gradient paths that can be trained memory-efficiently, thereby addressing issues (1) and (2). In the bigger picture, we believe that our approach of replacing gradient flow in HRNNs by locally computable losses may eventually help to attempt solving issue (3) as well.

## 2 RELATED WORK

Several techniques have been proposed to deal with the memory issues of TBPTT Chen et al. (2016); Gruslys et al. (2016). Specifically, they trade memory for computation by storing only certain hidden states and recomputing the missing ones on demand. This is orthogonal to our ideas and thus, both can potentially be combined.

The memory problem and the update lock have been tackled by online optimization algorithms such as Real Time Recurrent Learning (RTRL) Williams and Zipser (1989) and its recent approximations Tallec and Ollivier (2017); Mujika et al. (2018); Benzing et al. (2019); Cooijmans and Martens (2019). Online algorithms are promising, as the parameters can be updated in every step while the memory requirements do not grow with the sequence length. Yet, large computation costs and noise in the approximations make these algorithms impractical so far. Another way to deal with the parameter update lock and the memory requirements are Decoupled Neural Interfaces, introduced by by Jaderberg et al. in Jaderberg et al. (2017). Here, a neural network is trained to predict incoming gradients, which are then used instead of the true gradients.

HRNNs have been widely studied over the last decades. One early example by Schmidhuber in Schmidhuber (1992) proposes updating the upper hierarchy only if the lower one makes a prediction error. El Hihi and Bengio showed that HRNNs with fixed but different frequencies per level are superior in learning long-term dependencies El Hihi and Bengio (1996). More recently, Koutník et al. introduced the Clockwork RNN Koutnik et al. (2014), here the RNN is divided into several modules that are updated at exponentially decreasing frequencies. Many approaches have also been proposed that are explicitly provided with the hierarchical structure of the data Sordoni et al. (2015); Ling et al. (2015); Tai et al. (2015), for example character-level language models with word boundary information. Interestingly, Chung et al. Chung et al. (2016) present an architecture where this hierarchical structure is extracted by the network itself. As our model utilizes fixed or given hierarchical structure, yet does not learn it, models that can learn useful hierarchies may improve the performance of our approach.

Auxiliary losses in RNNs have been used to improve generalization or the length of the sequences that can be learnt. In Schmidhuber (1992), Schmidhuber presented an approach, where a RNN should not only predict a target output to solve the task, but also its next input as an auxiliary task. More recently, Goyal et al. showed that in a variational inference setting, training of the latent variables can be eased by an auxiliary loss which forces the latent variables to reconstruct the state of a recurrent network running backwards Goyal et al. (2017). Subsequently, Trinh et al. demonstrated that introducing auxiliary losses at random anchors in time which aim to reconstruct or predict previous or subsequent input subsequences significantly improve optimization and generalization of LSTMs Hochreiter and Schmidhuber (1997) and helps to learn time dependencies beyond the truncation horizon of TBPTT. The main difference to our approach is that they use auxiliary losses to incentivize an RNN to keep information in memory. This extends the sequences that can be solved by an additive factor through an echo-state network-like effect. On the contrary, we use auxiliary losses to replace gradient paths in hierarchical models. This, together with the right memorization scheme, allows us to discard all hidden states of the lower hierarchical levels when doing TBPTT, which reduces the memory requirements by a multiplicative factor.

## 3 METHODOLOGY

In this section we describe our approach in detail. We start by defining a standard HRNN, then explain how cutting its gradient flow saves memory during training, and conclude by introducing an auxiliary loss, that can be computed locally, which prevents a performance drop despite the restricted gradient flow (as shown in Section 4 below).

### 3.1 HIERARCHICAL RECURRENT NEURAL NETWORKS

The basis of our architecture is a simple HRNN with two hierarchical levels (deeper HRNNs are considered below). We describe it in terms of general RNNs. Such a RNN $X$ is simply a parameterized and differentiable function $f^X$ which maps a hidden state $h$ and an input $x$ to a new hidden state $h' = f^X(h, x)$.

The HRNN consists of the lower RNN (indicated by superscript $L$) and the upper RNN (superscript $U$). The lower RNN receives the input $x_t$ and generates the output $y_t$ in addition to the next hidden state $h_t$ at every time-step. Every $k$ steps, the upper RNN receives the hidden state of the lower RNN, updates its own hidden state and sends its hidden state to the lower RNN. Then, the hidden state of the lower RNN is reset to all zeros.

The update rules are summarized in Equation (1) for the lower and Equation (2) for the upper RNN. The unrolled computation graph is depicted in Figure 1.

$$h_t^L = \begin{cases} f^L(0, [x_t, h_t^U]) & \text{if} \quad t \mod k = 0, \\ f^L(h_{t-1}^L, [x_t, 0]) & \text{else.} \end{cases} \tag{1}$$

$$h_t^U = \begin{cases} f^U(h_{t-1}^U, h_{t-1}^L) & \text{if} \quad t \mod k = 0, \\ h_{t-1}^U & \text{else.} \end{cases} \tag{2}$$

While we have outlined the case for two hierarchical levels, this model naturally extends to deeper hierarchies by applying it recursively (see Section 4.4). This leads to levels that are updated exponentially less frequently. Moreover, the update frequencies can also be flexible in case the hierarchical structure of the input is explicitly provided (see Section 4.3).

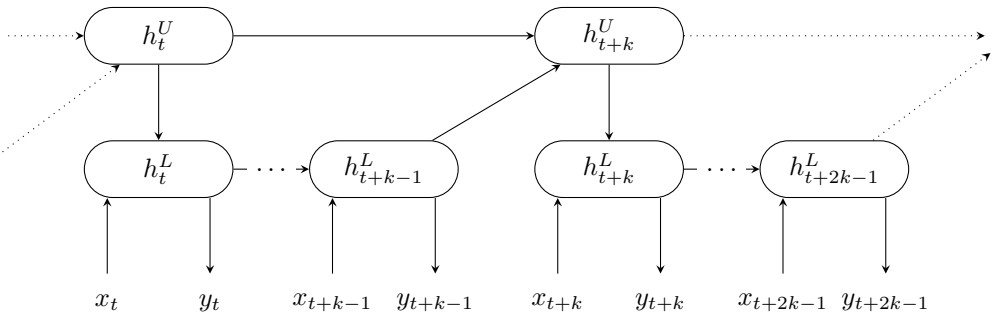

**Figure 1:** Unrolled computation graph of a HRNN with two hierarchical levels.

### 3.2 RESTRICTED GRADIENT FLOW

In TBPTT, the network is first unrolled for $T$ steps in the forward pass, see Figure 1, while the hidden states are stored in memory. Thereafter, the gradients of the loss with respect to the parameters are computed using the stored hidden states in the backward pass, see Figure 2 (a).

To save memory during gradient computation, we simply ignore the edges from higher to lower levels in the computation graph of the backward pass, see Figure 2. Importantly, the resulting gradients are not the true gradients and we term them *restricted gradients* as they are computed under restricted

gradient flow. We will refer to the *restricted gradients* with the $\tilde{\partial}$ symbol. Thus, $\frac{\tilde{\partial} h_t^U}{\partial h_{t-1}^L} = 0$ if $t$ is a time-step when the upper RNN ticks (i.e. $t \mod k = 0$) and it is equal to the true partial derivatives everywhere else. We call the HRNN that is trained using these restricted gradients the gradient restricted HRNN (**gr-HRNN**).

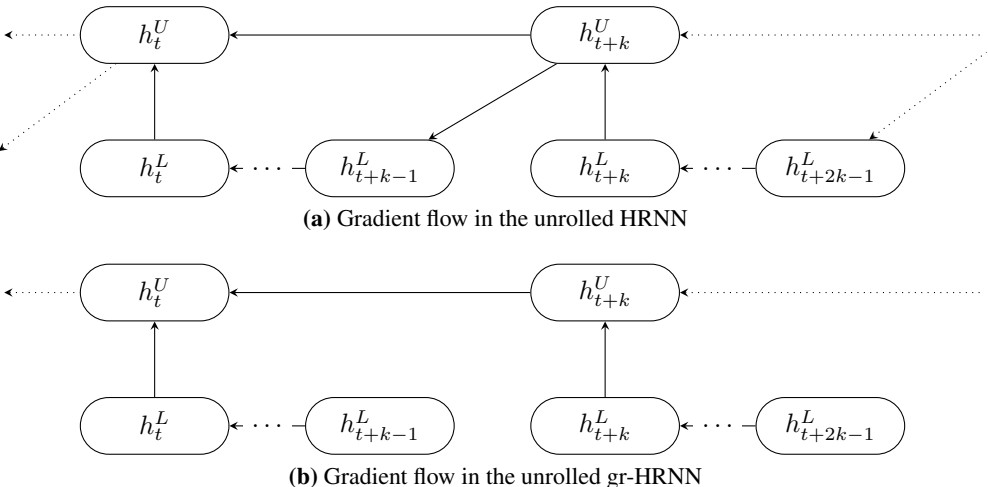

**(a)** Gradient flow in the unrolled HRNN

**(b)** Gradient flow in the unrolled gr-HRNN

**Figure 2:** Gradient flow in the unrestricted and gradient restricted HRNN.

Before we address the issue of compensating for the restricted gradients in the next section, we first analyze how much memory is needed to compute the restricted gradients.

In the following, let $t$ be a time-step when the upper RNN ticks (i.e. $t \mod k = 0$). A direct consequence of the restricted gradients is that $\sum_{i=t}^{T} \frac{\tilde{\partial} \mathcal{L}_i}{\partial h_{t-1}^L} = 0$, where $\mathcal{L}_i$ is the loss at time-step $i$ and we consider the network unrolled for time-steps $1, \ldots, T$. Therefore, right before the upper RNN updates, $\frac{\tilde{\partial} \mathcal{L}_i}{\partial \theta^L}$, where $\theta$ are the parameters of the network, can be computed for the previous $k$ time-steps using *just* the previous $k$ hidden states, $h_{t-k}^L, \ldots, h_{t-1}^L$, of the lower RNN. We further need these hidden states to compute $\sum_{i=1}^{k} \frac{\tilde{\partial} \mathcal{L}_{t-i}}{\partial h_{t-k}^U}$. However, we will not need the hidden states $h_{t-k}^L, \ldots, h_{t-1}^L$ for any other step of the backward pass and thus, we do not need to keep them in memory.

Therefore, the restricted gradients can be computed using memory proportional to $k$ for the hidden states $h_{t-k}^L, \ldots, h_{t-1}^L$ of the lower RNN and proportional to $2T/k$ for both the $T/k$ hidden states of the upper RNN and the $T/k$ accumulated restricted gradients of the loss with respect to the hidden states of the upper RNN. Standard TBPTT requires memory proportional to $T$ in order to compute the true gradients. Here, we showed that the restricted gradients can be computed using memory proportional to $k + 2T/k$, thus improving by a factor of $2/k$. For (deep) HRNNs with $l$ levels, the above argument can be applied recursively, which yields memory requirements of $(l-1)k$ for the hidden states of the lower RNNs and $2T/k^{l-1}$ for the uppermost RNN.

### 3.3 LOCALLY COMPUTABLE AUXILIARY LOSSES

In the gr-HRNN, the lower RNN is not incentivized to provide a useful representation as input for the upper RNN. However, it is clear that if this hidden state which is sent upwards contains all information about the last $k$ inputs, then the upper RNN has access to all information about the input, and thus the gr-HRNN should be able to learn as effectively as the HRNN without gradient restrictions.

Hence, we introduce an auxiliary loss term to force this hidden state to contain all information about the last $k$ inputs. This auxiliary loss comes from a simple feed-forward decoder network, which given this hidden state and a randomly sampled index $i \in \{1, \ldots, k\}$ (in one-hot encoding) must output the $i$-th previous input. Then, the combined loss of the HRNN is the sum of the loss defined by the task

and the auxiliary decoder loss multiplied by a hyper-parameter $\beta$. For hierarchies with $l$ levels, we add $l - 1$ decoder networks with respectively weighted losses.

Crucially, the auxiliary loss can be computed locally in time. Thus, the memory requirements for training increase only by an additive term in the size of the decoder network.

## 4 EXPERIMENTS

In this section, we experimentally test replacing the gradient flow from higher to lower levels by adding the local auxiliary losses. Therefore, we evaluate the gr-HRNN with auxiliary loss (simply termed 'ours' from here on) on four different tasks: Copy task, pixel MNIST classification, permuted pixel MNIST classification, and character-level language modeling. All these tasks require that the model effectively learns long-term dependencies to achieve good performance. Additionally, to our model, we also evaluate several ablations/augmentations of it to fully understand the importance and effects of its individual components. These other models are:

- **HRNN**: This model is identical to ours, except that all gradients from higher to lower levels are propagated. The weight of the auxiliary loss is a hyper-parameter which is set using a grid-search (as the auxiliary loss may improve the performance, we keep it here to make a fair comparison with our model). While this model has a much larger memory cost for training, it provides an upper bound on the performance we can expect from our model.

- **gr-HRNN**: This model is equivalent to our model, except that the weight of the auxiliary loss is set to zero. Thereby, it permits studying if using the auxiliary loss function (and the corresponding decoder network) is really necessary when the gradients are stopped.

- **mr-HRNN**: This model is identical to the **HRNN**, except that it is trained using only as much memory as our model requires for training. That is, it is unrolled for a much smaller number of steps than the other models. In consequence, it admits a fair performance comparison (in terms of memory) to our model.

If not mentioned otherwise below, the parameters are as follows. All RNNs are LSTMs with a hidden state of 256 units. The network for the auxiliary loss is a two layer network with 256 units, a ReLU Glorot et al. (2011) non-linearity and uses either the cross-entropy loss for discrete values or the mean squared loss for continuous ones. For each model we pick the optimal $\beta$ from $[10^{-3}, 10^{-2}, 10^{-1}, 0, 10^0, 10^1, 10^2]$, except for the **gr-HRNN** for which $\beta = 0$. The models are trained using a batch size of 100 and the Adam optimizer Kingma and Ba (2014) with the default Tensorflow Abadi et al. (2015) parameters: learning rate of 0.001, $\beta_1 = 0.9$ and $\beta_2 = 0.999$.

### 4.1 COPY TASK

**Table 1:** Results for the copy task (3 repetitions). Memory refers to the number of vectors of the same size as the hidden state that need to be stored for each algorithm and batch element.

| Model | Gradients | Aux. loss | Memory | $L_{max}$ (longest solved sequences) |
|---|---|---|---|---|
| ours | ✗ | ✓ | 50 | $108 \pm 0$ |
| gr-HRNN | ✗ | ✗ | 50 | $21 \pm 13$ |
| HRNN | ✓ | ✓ | 200 | $107.7 \pm 0.47$ |
| mr-HRNN | ✓ | ✓ | 50 | $48.3 \pm 0.94$ |

In the copy task, the network is presented with a binary string and should afterwards output the presented string in the same order (an example for a sequence of length 5 is: input `01101*****` with target output `*****01101`). We refer to the maximum sequence length that a model can solve (i.e. achieves loss less than 0.15 bits/char) as $L_{max}$. Table 1 summarizes the results for a sequence of length 100.

The copy task requires exact storage of the input sequence over many steps. Since the length of the sequence and therefore the dependencies are a controllable parameter in this task, it permits to explicitly assess how long dependencies a model can capture.

Both our model and the model using all gradients (**HRNN**) achieve very similar performance, which is limited by the number of steps for which the network is unrolled in training. Moreover, the auxiliary loss is necessary, as the model without it (**gr-HRNN**) performs poorly. Moreover, our model drastically outperforms the model with all gradients, when given the same memory budget, as the memory restricted HRNN (**mr-HRNN**) also performs poorly. Notably, our model is remarkably robust to the choice of the auxiliary loss weight $\beta$, as for all values except 0 it learns sequences of length at least 100. Finally, the **HRNN** actually uses the capacity of the decoder network, as the grid-search yields a nonzero $\beta$.

All models, except the **mr-HRNN**, are unrolled for $T = 200$ steps and the upper RNN ticks every $k = 10$ steps of the lower one. To equate memory budgets, the **mr-HRNN** is unrolled for $2T/k + k = 50$ steps. Hence, for sequence lengths at most 100, gradients are propagated through the whole sequence. The model parameters are only updated once per batch.

## 4.2 (Permuted) Pixel MNIST Classification

**Table 2:** Results for the pixel MNIST tasks (3 repetitions). Memory refers to the number of vectors of the same size as the hidden state that need to be stored for each algorithm and batch element.

| Model | Gradients | Aux. loss | Memory | Accuracy | Accuracy (permuted) |
|---|---|---|---|---|---|
| ours | ✗ | ✓ | 166 | $0.9886 \pm 0.0002$ | $0.9680 \pm 0.0008$ |
| gr-HRNN | ✗ | ✗ | 166 | $0.2109 \pm 0.0743$ | $0.9368 \pm 0.0049$ |
| HRNN | ✓ | ✓ | 784 | $0.9885 \pm 0.0003$ | $0.9681 \pm 0.0008$ |
| mr-HRNN | ✓ | ✓ | 166 | $0.8939 \pm 0.0041$ | $0.9553 \pm 0.0015$ |

In the pixel MNIST classification task, the pixels of MNIST digits are presented sequentially. After observing the whole sequence the network must predict the class label of the presented digit. The permuted pixel MNIST task is exactly the same, but the pixels are presented in a fixed random order. Table 2 summarizes the accuracy of the different models on the two tasks.

Both pixel MNIST classification tasks require learning long-term dependencies, especially when using the default permutation, as the most informative pixels are around the center of the image (i.e. in the middle of the input sequence) and the class prediction is only made at the end. Additionally, unlike in the copy task, the input has to be processed in a more complex manner in order to make a class prediction.

The results on both tasks are in line with the copy task. Our model performs on par with the model using the true gradient (**HRNN**). Again, the auxiliary loss is necessary, as the accuracy of the **gr-HRNN** is significantly worse. Given the same memory budget, our model outperforms the HRNN with all gradients.

All models are unrolled for $T = 784$ steps (the number of pixels of an MNIST image) except for the memory restricted model which is unrolled for 166 steps. The upper RNN ticks every $k = 10$ steps of the lower one. The same permutation is used for all runs of the permuted task.

## 4.3 Character-level Language Modeling

In the character-level language modeling task, a text is presented character by character and at each step, the network must predict the next character. The results for this task on the Penn TreeBank corpus Marcus and Marcinkiewicz (1993) are summarized in Table 3.

In contrast to the previous tasks, character-level language modeling contains a complex mix of both short- and long-term dependencies, where short-term dependencies typically dominate long-term

**Table 3:** Results for the character-level language modeling task on the Penn TreeBank corpus (3 repetitions).

| Model | Gradients | Aux. loss | Validation bits/char | Test bits/char |
|---|---|---|---|---|
| ours | ✗ | ✓ | $1.4519 \pm 0.0014$ | $1.4027 \pm 0.0003$ |
| gr-HRNN | ✗ | ✗ | $1.4545 \pm 0.0016$ | $1.4074 \pm 0.0011$ |
| HRNN | ✓ | ✓ | $1.4480 \pm 0.0011$ | $1.3974 \pm 0.0021$ |

dependencies. Thus, one may expect that replacing the true gradients by the local loss is the most harmful here.

The performance of all 3 models is very close in this task, even for the **gr-HRNN**, which has neither gradients nor the auxiliary loss to capture long-term dependencies. We believe that this is due to the dominance of short-term dependencies in character-level language modeling. It has been widely reported that the length of the unrolling, and thus long-term dependencies, have a minimal effect on the final performance of character-level language models Jaderberg et al. (2017), particularly for small RNNs, as in our experiments. Still, we observe the best performance when using the true gradients (**HRNN**), a slight degradation when replacing gradients by the auxiliary loss (our), which slightly improves on not using the auxiliary loss (**gr-HRNN**).

Here, we explicitly provide the hierarchical structure of the data to the model by updating the upper RNN once per word, while the lower RNN ticks once per character. As discussed in Section 2, there are models that can extract the hierarchical structure, which is a separate goal from ours.

We refrain from comparing with a memory restricted model and displaying memory budgets because the dynamic memory requirements depending on the input sequence prohibit a fair comparison. However, as stated earlier, the unroll length usually has a minimal effect on the performance of character-level language modeling. We unroll the models for 50 characters and use an upper RNN with 512 units to deal with the extra complexity of the task.

## 4.4 DEEPER HIERARCHIES

In deeper hierarchies, the output modalities of the individual decoder networks are different: whereas the decoder network in the lowest level has to predict elements of the input sequence (e.g. bits in the copy task), the decoder networks in higher levels have to predict hidden states of the lower level's RNNs. Here, we confirm that our approach generalizes to deeper hierarchies.

In particular, we use the copy task with sequence length 500 (and truncation horizon 1000) to test this. We consider a HRNN with 3 levels, where the lowest RNN is updated in every step, the middle RNN every 5 steps and the upper RNN every 25 steps, where no gradients from higher to lower RNNs are propagated. We compare using the auxiliary loss only on the lowest level with using it in the lowest and the middle layer. Thereby, we check whether applying the auxiliary loss in the middle RNN (i.e. over hidden states rather than raw inputs) is necessary to solving the task.

The RNNs have 64, 256 and 1024 units (from lower to higher levels). The $\beta$ for the lowest level is 0.1 and 1 for the middle level. All other experimental details are kept as in Section 4.1. We run several repetitions which are shown in Figure 3. Without the auxiliary loss in the middle layer, the network cannot solve the task (in fact, not even close as performance is close to chance level). However, when the auxiliary loss is added also to the middle layer, the model can solve the task perfectly on every run.

## 5 CONCLUSION

In this paper, we have shown that in hierarchical RNNs the gradient flow from higher to lower levels can be effectively replaced by locally computable losses. This allows memory savings up to an exponential factor in the depth of the hierarchy. In particular, we first explained how not propagating gradients from higher to lower levels permits these memory savings. Then, we introduced auxiliary losses that encourage information to flow up the hierarchy. Finally, we demonstrated experimentally

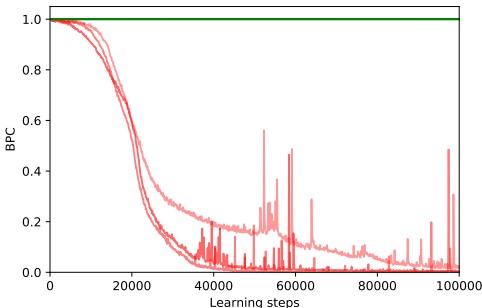

**Figure 3:** Results on copy task with a sequence length of 500. Three different runs of deep HRNNs with the auxiliary loss in both levels (red) and with the auxiliary loss only in the lower level (green).

that the memory-efficient HRNNs with our auxiliary loss perform on par with the memory-heavy HRNNs and strongly outperform HRNNs given the same memory budget on a wide range of tasks, including deeper hierarchies.

High capacity RNNs, like Differentiable Plasticity Miconi et al. (2018), or Neural Turing Machines Graves et al. (2014) have been shown to be useful and even achieve state-of-the-art in many tasks. However, due to the memory cost of TBPTT, training such models is often impractical for long sequences. We think that combining these models with our techniques in future work could open the possibility for using high capacity RNNs for tasks involving long-term dependencies that have been out of reach so far.

Still, the problem of the parameter update lock remains. While this is the most under-explored of the three big problems when training RNNs (vanishing/exploding gradients and memory requirements being the other two), resolving it is just as important in order to be able to learn long-term dependencies. We believe that the techniques laid out in this work (i.e. replacing gradients in HRNNs by locally computable losses) can be a stepping stone towards solving the parameter update lock. We leave this for future work.

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
