# OpenReview forum: "Decoupling Hierarchical Recurrent Neural Networks With Locally Computable Losses"
_ICLR.cc/2020/Conference — Reject_

### Official Review · AnonReviewer2 · 2019-10-09
**Official Blind Review #2**

**Rating:** 1

**Review:**

Summary:
The paper introduces a hierarchical RNN architecture that could be trained more (memory) efficiently. The difference in the architecture seems to be an auxiliary loss that decodes k step inputs and some perturbation of TBPTT.

Comments on the paper

1. The paper seems to be have been written in a rush. The language could be improved, the format is not always consistent and in general the paper could be much better written. There are quite some typos as well in the paper, for example , Trinh et al.  is not a proper citation.

2. The authors mentioned that TBPTT is not memory efficient, this is not very clear to me, as it only needs to keep the number of truncation steps that it backprops through and hence much more memory efficient compared to full BPTT.

3. It is not clear to me what is the benefit of gr-HMRNN. It is not clear why cutting of the gradients from the higher level to the lower level would help.

4. It is surprising to me that HMRNN could only solve the copy task upto a length of 108.

5. I would also suggest another copy task from Hochreiter, Sepp and Schmidhuber, Jürgen. Long short-term memory. Neural computation, 9(8): 1735–1780, 1997.

In general, the paper seems to have been written in a rush. I would recommend the papers to be revised.

**Experience Assessment:**

I have published one or two papers in this area.

**Review Assessment: Checking Correctness Of Derivations And Theory:**

I assessed the sensibility of the derivations and theory.

**Review Assessment: Checking Correctness Of Experiments:**

I assessed the sensibility of the experiments.

**Review Assessment: Thoroughness In Paper Reading:**

I read the paper at least twice and used my best judgement in assessing the paper.

---

### Official Review · AnonReviewer1 · 2019-10-20
**Official Blind Review #1**

**Rating:** 1

**Review:**

The proposed work investigates the problem of learning hierarchy in RNNs. Authors note that different layers of the hierarchy are trained in "sync". The proposed paper suggests to decouple the different layers of hierarchy using auxiliary losses.  The form of auxiliary losses used in the paper are of the form of local losses, where there is a decoder, which is used to decode past inputs to each level from the hidden state that is sent up the hierarchy, therebyforcing this hidden state to contain all relevant information.

Clarity of the paper: The paper is clearly written.

Method: The proposed method  ignores the gradients from higher to lower levelsin the backward pass,  (because of this, the authors can also save some memory). In order to compensate for the lost gradients, authors propose to use local losses, and we introduce an auxiliary loss term to force this hidden state to contain all information aboutthe last k inputs. The authors note that the hidden state from the lower level (to the higher level) should contain the summary of the past, and hence use a decoder network (which is simply parameterized) as a feedforward network which is used to decoder a "past" hidden state.

Related Work Section: The related work section is nicely written. The authors have covered mostly everything. These 3 papers may still be relevant. (a), (b), (c).   (b) could be relevant for mitigating the parameter update lock problem as mentioned by authors in the introduction of the paper. (c) is also relevant as authors in (c) also consider using auxiliary losses for learning long term dependencies.
(a) SkipRNN: https://arxiv.org/abs/1708.06834(b) Sparse Attentive Backtracking: http://papers.nips.cc/paper/7991-sparse-attentive-backtracking-temporal-credit-assignment-through-reminding
(c)  Learning long term dependencies in RNNs using auxiliary losses https://arxiv.org/abs/1803.00144
Experiment Section: In order to validate the proposed method, authors evaluate it on copying task, pixel MNIST classification, permutedpixel MNIST classification, and character-level language modeling.
a) Copying results show that the decoder network are essential to achieve decent results. This task though does not show the strength of the proposed method though as baseline also solves the problem completely. It might be interesting to actually scale the "gap" time in copying time step to something larger like T = 1000 or something.
b) PIXEL MNIST classification: Authors use the pixel by pixel classification task to test the proposed method. Here, the proposed method performs comparable to the hierarchical RNN (but without using too much memory).
c) Character level modelling: Authors demonstrate the performance of the proposed method on language modelling task (PTB). These results are particularly not interesting, as the performance gain is very marginal. Also, may be using other language modelling datasets like Wikitest103 or Text8 might be more useful.  As for the results, even unregularized LSTM performs better than the baseline in this paper. (For reference, see https://arxiv.org/abs/1606.01305)

What authors can do to improve paper:
- The problem considered in the proposed paper is very interesting to me. Though, the results are not (yet) convincing. It might be interesting to think about a task, where there are really long term dependencies like reading CIFAR10 digit pixel by pixel and then doing classification, where the authors can actually show the promise of the proposed method.
- It might also be interesting to know how are the original training cost objective is weighed against the auxiliary loss. Have authors tried any search over what kind of auxiliary loss performs well ?

**Experience Assessment:**

I have published in this field for several years.

**Review Assessment: Checking Correctness Of Derivations And Theory:**

I carefully checked the derivations and theory.

**Review Assessment: Checking Correctness Of Experiments:**

I carefully checked the experiments.

**Review Assessment: Thoroughness In Paper Reading:**

I read the paper thoroughly.

---

### Official Review · AnonReviewer3 · 2019-11-09
**Official Blind Review #3**

**Rating:** 1

**Review:**

Claim: Backpropagation of gradients from a higher to lower level in a HRNN can be removed and replaced with auxiliary losses predicting input tokens at the lower level without affecting performance.

Significance: The significance of the claim hinges on whether HRNNs are more effective than other methods designed to help RNNs capture long-term dependencies (e.g. stacking RNNs or using different architectures). I think the authors could make a more substantive argument why this would be the case in the introduction, but they do a nice job of situating their work in the context of the present literature.

Novelty: The proposed method is not very original, since augmenting RNNs with auxiliary losses in order to better capture long-term dependencies has been used in many previous papers. The authors mention some of these papers in the related work section.

Clarity: The paper's description of the proposed method is well-written. Some parts of the experiment section could be made clearer.
--  I encourage the authors to invent a new acronym to refer to "our model" (perhaps aux-HRNN?). In the description of the mr-HRNN (pg. 5), I find the sentence "trained using only as much memory as our model requires for training" confusing.  I initially thought our model referred to the mr-HRNN in the setence.
-- Training settings (e.g. the number of ticks of the upper RNN) should be described at the beginning of each section.
-- A seeming contradiction is made when discussing the results in 4.3. First, it said that because short term dependencies dominate long term dependencies it is expected that the proposed method will suffer greatly (pg. 6, bottom). In the next paragraph, it is claimed that all three models perform similarly due to the same reason. Which is it?

Supporting evidence: The claim is empirical and the supporting evidence is experimental. As such, I find the comprehensiveness of the experiments wanting. There are several ways the experiments could be improved.
-- Results for each \beta value should be included, to see how placing increasing significance on the auxiliary loss impacts the results.
-- Include all relevant details necessary to reproduce the results, such as the length of training or stopping criterion used.
-- Additional results when varying the number of ticks.
-- More results with deeper hierarchies, since the ability to capture salient information at different levels of coarseness is a key selling point of HRNNs.
-- Results on larger scale tasks besides character level language modelling on Penn TreeBank.

Other comments:
-- In the intro, I think some mention of parallel architectures such as transformers or convolutional architectures is warranted here, since parallelizability of training is a significant reason why these architectures are becoming preferred over RNNs.
-- Citations are mishandled throughout the paper. Citations should be enclosed in parentheses unless used as a subject in the sentence (e.g. "Sordoni et al. make the case that..."). There is no need to refer to a citation twice in a sentence, like you do in "More recently, Koutnik et al. introduced the Clockwork RNN Koutnik et al. (2014)..."
-- I don't understand why the permuted accuracy of the gr-HRNN is so much higher than the non-permuted accuracy. One possible explanation is that the important pixels ended up at the end in each of the three trials, hence the gr-HRNN did not have to remember much information from the past. This should be addressed in the paper.
-- I would welcome some theoretical analysis as to why replacing the gradient path with this particular auxiliary loss does not impact results. I also think some discussion of what this means HRNNs are actually doing might be nice as well.

**Experience Assessment:**

I have published one or two papers in this area.

**Review Assessment: Checking Correctness Of Derivations And Theory:**

I carefully checked the derivations and theory.

**Review Assessment: Checking Correctness Of Experiments:**

I carefully checked the experiments.

**Review Assessment: Thoroughness In Paper Reading:**

I read the paper thoroughly.

---

### Author Response · Authors · 2019-11-15
**Reply**

We thank the reviewers for their thoughtful comments. Due to the low scores, we decided to not update our manuscript but we will still include the useful feedback into future revisions of the paper.

---

### Decision · Program_Chairs · 2019-12-19

**Decision:**

Reject

**Comment:**

All reviewers gave this paper a score of 1.
The AC recommends rejection.